# Discoidin Domain Receptor 1 Regulates Runx2 during Osteogenesis of Osteoblasts and Promotes Bone Ossification via Phosphorylation of p38

**DOI:** 10.3390/ijms21197210

**Published:** 2020-09-29

**Authors:** Liang-Yin Chou, Chung-Hwan Chen, Shu-Chun Chuang, Tsung-Lin Cheng, Yi-Hsiung Lin, Hsin-Chiao Chou, Yin-Chih Fu, Yan-Hsiung Wang, Chau-Zen Wang

**Affiliations:** 1Graduate Institute of Medicine, College of Medicine, Kaohsiung Medical University, Kaohsiung 80708, Taiwan; laining59@gmail.com (L.-Y.C.); rna.studio2014@gmail.com (H.-C.C.); microfu@gmail.com (Y.-C.F.); 2Orthopaedic Research Centre, Kaohsiung Medical University, Kaohsiung 80708, Taiwan; hwan@kmu.edu.tw (C.-H.C.); hawayana@gmail.com (S.-C.C.); junglecc@gmail.com (T.-L.C.); yhwang@kmu.edu.tw (Y.-H.W.); 3Regeneration Medicine and Cell Therapy Research Center, Kaohsiung Medical University, Kaohsiung 80708, Taiwan; 4Department of Orthopedics, College of Medicine, Kaohsiung Medical University, Kaohsiung 80708, Taiwan; 5Department of Orthopedics, Kaohsiung Municipal Ta-Tung Hospital, Kaohsiung Medical University, Kaohsiung 80145, Taiwan; 6Institute of Medical Science and Technology, National Sun Yat-Sen University, Kaohsiung 80424, Taiwan; 7Division of Adult Reconstruction Surgery, Department of Orthopedics, Kaohsiung Medical University Hospital, Kaohsiung Medical University, Kaohsiung 80708, Taiwan; 8Cardiovascular Research Centre, College of Medicine, National Cheng Kung University, Tainan City 70101, Taiwan; 9Department of Physiology, College of Medicine, Kaohsiung Medical University, Kaohsiung 80708, Taiwan; 10Department of Medical Research, Kaohsiung Medical University Hospital, Kaohsiung 80708, Taiwan; 11Department of Biotechnology, Kaohsiung Medical University, Kaohsiung 80708, Taiwan; caminolin@gmail.com; 12Division of Cardiology, Department of Internal Medicine, Kaohsiung Medical University Hospital, Kaohsiung 80756, Taiwan; 13Lipid Science and Aging Research Center, Kaohsiung Medical University, Kaohsiung 80708, Taiwan; 14School of Dentistry, Kaohsiung Medical University, Kaohsiung 80708, Taiwan

**Keywords:** bone, discoidin domain receptor 1, DDR1, osteoblast, osteogenesis, bone development

## Abstract

Discoidin domain receptor 1 (*Drd1*) is a collagen-binding membrane protein, but its role in osteoblasts during osteogenesis remains undefined. We generated inducible osteoblast-specific *Ddr1* knockout (OKOΔ*Ddr1*) mice; their stature at birth, body weight and body length were significantly decreased compared with those of control *Ddr1^f/f-4OHT^* mice. We hypothesize that *Ddr1* regulates osteogenesis of osteoblasts. Micro-CT showed that compared to 4-week-old *Ddr1^f/f-4OHT^* mice, OKOΔ*Ddr1* mice presented significant decreases in cancellous bone volume and trabecular number and significant increases in trabecular separation. The cortical bone volume was decreased in OKOΔ*Ddr1* mice, resulting in decreased mechanical properties of femurs compared with those of *Ddr1^f/f-4OHT^* mice. In femurs of 4-week-old OKOΔ*Ddr1* mice, H&E staining showed fewer osteocytes and decreased cortical bone thickness than *Ddr1^f/f-4OHT^*. Osteoblast differentiation markers, including BMP2, Runx2, alkaline phosphatase (ALP), Col-I and OC, were decreased compared with those of control mice. *Ddr1* knockdown in osteoblasts resulted in decreased mineralization, ALP activity, phosphorylated p38 and protein levels of BMP2, Runx2, ALP, Col-I and OC during osteogenesis. Overexpression and knockdown of *Ddr1* in osteoblasts demonstrated that DDR1 mediates the expression and activity of Runx2 and the downstream osteogenesis markers during osteogenesis through regulation of p38 phosphorylation.

## 1. Introduction

Bone development is a dynamic process within higher organisms. In mammals, skeletogenesis is a two-step ossification process comprising endochondral ossification and intramembranous ossification. The endochondral ossification process starts with mesenchymal stem cells (MSCs) committing to chondrocytes, in which proliferation, differentiation, hypertrophy, terminal differentiation and apoptosis result in the formation of bone; osteoblasts and osteoclasts comprise the majority of cells in mature bone [1]. By contrast, in the intramembranous ossification process, MSCs in cortical and trabecular bone are directly differentiated into osteoblasts [2] to form flat bone structures, including the skull, clavicle, jaw, sternum, ribs, and collar bone. At the periosteum, osteoprogenitor cells differentiate into osteoblasts and are responsible for mineralization of the bone matrix during bone formation. Runx2 (Runt-related transcription factor 2, also known as Cbf1) is a key regulator during osteoblast differentiation, which induces preosteoblasts to become mature osteoblasts. Specific deletion of Runx2 in osteoblasts leads to defective intramembranous ossification and postnatal bone formation [3,4]. Runx2 regulates the expression of genes at different stages of osteogenesis, such as the early stage during which preosteoblasts commit to osteoblasts by secreting alkaline phosphatase (ALP) and type I collagen (Col-I) and expressing the differentiation marker osteopontin (OPN) and the later stage during which the mature osteoblast markers osteocalcin (OC), which is a calcium binding protein, bone sialo protein (BSP) and matrix metalloproteinase-13 (MMP13) are expressed to promote mineralization [3,4,5,6,7,8,9]. The basic condition of bone growth is the dynamic balance of cartilage and cortical bone formation.

Discoidin domain receptors (DDRs) are a family of receptor tyrosine kinases that includes DDR1 and DDR2, which are transmembrane collagen-binding receptors. DDR1 is commonly expressed in a variety of mice and in humans, and DDR1 gene maps indicate 17 exons at chromosome 6 (6p21.3) in the human genome, which can be expressed as five distinct DDR1 isoforms. However, DDR2 gene maps show 19 exons at chromosome 1 (1q23.3) but only one isoform [10,11]. Fibrillar collagens are ligands for both DDRs; thus far, DDR1 and DDR2 could be activated by collagen, including types I−III, V and X [11,12,13]. In particular, DDR1 also binds type VIII collagen and basement membrane type IV collagen [11]. Global DDR1 knockout in mice causes dwarfism, and the body weight is more decreased than in wild-type mice. Micro-CT results show that the global DDR1 knockout mice are short and thinner in fibular bone but the growth plate length, chondrocyte proliferation and apoptosis are unaltered compare with wild-type mice [10]. However, the mechanism of global *Ddr1* knockout that causes dwarfism in mice and whether DDR1 regulates the function of osteoblasts during development remain undefined. DDR1 is also expressed at the periosteal collar of the clavicle in embryonic and postnatal mice [10]. Global *Ddr1*-null mice have abnormal development in multiple organ systems, which may affect the interpretation of DDR1 function, such as loss of auditory function, lactation impairment, cochlear duct defect, reproduction defects, and proteinuria [11,12,13,14,15]. There are no reports that show the role of Ddr1 in osteoblast function during osteogenesis. Therefore, we firstly created the Ddr1f/f mice, and used the Cre-LoxP system to generate the inducible osteoblast-specific Ddr1 knockout mice (OKOΔDdr1) in this study. Our results showed that there were no other developmental abnormalities compared with global Ddr1 knockout mice. The OKOΔ*Ddr1* mice showed a shorter stature at birth than the control Ddr1^f/f-4OHT^ mice. Therefore, we hypothesize that Ddr1 may play a role in regulating the osteogenesis of osteoblasts. To elucidate the role of *Ddr1* in regulating osteoblasts during bone development, we used micro-CT analysis, H&E staining and IHC staining to evaluate the expression of osteoblast differentiation markers in the femurs in the OKOΔDdr1 mice in vivo. We also used an in vitro model of Ddr1 knockdown in osteoblasts, which showed decreased mineralization, ALP activity, levels of phosphorylated p38 and protein levels of BMP2, Runx2, ALP, Col-I and OC during osteogenesis. In this study, by using overexpression and knockdown of Ddr1, we demonstrated that, via p38, DDR1 mediates the expression and activity of Runx2 and the downstream osteogenesis markers OC, Col-I, and ALP during osteogenesis.

## 2. Results

### 2.1. Generation of Osteoblast-Specific Knockout Mice (a1(I) Collagen-CreERT; Ddr1^f/f^; OKOΔDdr1 Mice)

To investigate the role of *Ddr1* in osteoblasts during bone ossification, we first generated 4-OHT inducible osteoblast-specific *Ddr1* knockout (a1(I)-CreERT; *Ddr1*^f/f^) mice on the OKO background. To confirm the osteoblast-specific knockout of *Ddr1* in OKOΔ*Ddr1* mice, we conducted gene expression, Western blot, and IHC staining assays in *Ddr1^f/f-4OHT^* and OKOΔ*Ddr1* mice. The qPCR results showed that the gene expression of *Ddr1* in OKOΔ*Ddr1* mice was significantly decreased by approximately 95% compared with that in *Ddr1^f/f-4OHT^* mice (Figure 1A). Additionally, the protein level of DDR1 was decreased by approximately 90% in OKOΔ*Ddr1* mice compared with *Ddr1^f/f-4OHT^* mice (Figure 1B). To further demonstrate that we specifically knocked out *Ddr1* in osteoblasts, we performed IHC staining for DDR1 in the femurs of 4-week-old mice, the results of which showed that DDR1 staining was obvious in the cortical bone, periosteum and articular chondrocytes in *Ddr1^f/-4OHT^* mice. However, DDR1 staining was decreased by approximately 90% in cortical bone in OKOΔ*Ddr1* mice but was still observed in articular chondrocytes (Figure 1C,D). By contrast, DDR2 staining in cortical bone and articular chondrocytes was similar in *Ddr1^f/f-4OHT^* and OKOΔ*Ddr1* mice and showed no significant quantitative difference (Figure 1C,D). These results indicated that we had succeeded in the creation of osteoblast-specific *Ddr1* knockout (OKOΔ*Ddr1*) and that this model did not affect the expression of *Ddr2*, suggesting that there is no compensatory effect of *Ddr2* expression. We stained the osteoclasts by TRAP staining, and also immunochemistry stain of TRAcP as an osteoclast marker. The results showed that knockout *Ddr1* in osteoblasts had no effect on osteoclast activity and the expression of osteoblasts (S1A, 1B). These results indicated that knockout *Ddr1* in osteoblasts would not influence the osteoclast phenotype, and demonstrated the osteoblast-specific Ddr1 knockout in mice.

### 2.2. Skeletal Dysplasia in OKOΔDdr1 Mice

To study the role of *Ddr1* in osteoblasts during postnatal skeletal development, we performed double staining of the skeleton with Alizarin red for mineralized matrix (red) and Alcian blue for cartilaginous matrix (blue). Although selectively knocking out *Ddr1* in osteoblasts was not embryonic lethal and did not cause structurally significant abnormalities in OKOΔ*Ddr1* mice, the long bones of OKOΔ*Ddr1* mice were thinner than those of *Ddr1^f/f-4OHT^* mice (Figure 2A). Knocking out *Ddr1* in osteoblasts caused a decrease in the mineralization of the rib/sternum, which is driven by intramembranous ossification to form bones (Figure 2B). We also observed that the length and diameter of endochondral ossification bones (such as forelimb, hindlimb, and vertebrae) were smaller in OKOΔ*Ddr1* mice than in *Ddr1^f/f-4OHT^* mice (Figure 2C). At 4 weeks of age, OKOΔ*Ddr1* mice exhibited significantly decreased body weight and body length compared to those of *Ddr1^f/f-4OHT^* mice (Figure 2D). These findings indicate that Ddr1 knockout in osteoblasts causes dysplasia, which suggests that DDR1 plays a critical role in osteoblast function during ossification.

### 2.3. Decreased Bone Formation in the Tibia of OKOΔDdr1 Mice

At the end of endochondral ossification, hypertrophic cartilage is replaced by osteoblasts, osteoclasts, and blood vessels. At the perichondrium of cortical bone, osteoprogenitor cells differentiate into osteoblasts and form the bone collar of the cartilage anlage. In 2- and 4-week-old OKOΔ*Ddr1* mice, we observed that the long bones of the femur and tibia were slimmer than those in *Ddr1^f/f-4OHT^* mice (Figure 3A). To characterize how *Ddr1* regulates osteoblast function during skeletal development, we performed micro-CT analysis. The reconstructed 3-D images revealed that the long bones were obviously thicker in *Ddr1^f/f-4OHT^* mice than in OKOΔ*Ddr1* mice (Figure 3B). The cross-sectional views and 3-D reconstruction of the trabecular bone (2-mm segment) showed less trabecular bone in *Ddr1^f/f-4OHT^* mice than in OKOΔ*Ddr1* mouse tibia (Figure 3C). Moreover, the 3-D structure of tibial cortical bone was thinner and smaller in *Ddr1^f/f-4OHT^* mice than in OKOΔ*Ddr1* mice (Figure 3C). To investigate whether *Ddr1* knockout in osteoblasts affects bone mineralization in vivo, we analyzed the 3-D structure of tibias (2 mm length) in 4-week-old mice, and the inner region for trabecular bone and outer region for cortical bone were also analyzed. The trabecular bone volume (BV/TV) and trabecular number (Tb.N) were decreased and the trabecular separation (Tb.Sp) was increased in OKOΔ*Ddr1* mice (Figure 3D). It had no significant differences on trabecular bone mineral density (Tb. BMD) and trabecular thickness (Tb.Th) (Figure 3D). The results of the cortical bone quantitation showed a significant decrease in the bone volume and bone mineral content (BMC) but no effect on the bone mineral density (BMD) in OKOΔ*Ddr1* mice (Figure 3E). Consistent with the micro-CT data, 4-week-old OKOΔ*Ddr1* mice showed a marked reduction in bone structure properties (mechanism loading to failure) such as max-load, break point, stiffness, area under the max curve, area under the break curve, modulus, and toughness in femur cortical bone compared to those in *Ddr1^f/f-4OHT^* mice; these properties were assessed with the three-point bending analysis (Table 1). These results indicated that osteoblasts with *Ddr1* knockout caused the bone structure to be significantly weaker than that of the control mice. Taken together, these results indicate that osteoblasts with *Ddr1* knockout decrease the function of osteoblasts and decrease bone formation of cortical bone and trabecular bone, leading to bone ossification delay.

### 2.4. Knocking out Ddr1 in Osteoblasts Decreases Cortical Bone Thickness and Cell Number

To further understand the role of Ddr1-mediated osteoblast function and osteogenesis-related markers during bone ossification, we performed Safranin O/fast green and hematoxylin and eosin (H&E) staining. In the Safranin O/fast green staining assay, trabecular bone was less abundant in 4-week-old OKOΔ*Ddr1* mouse femurs (Figure 4A), but quantification of the growth plate length showed no difference between *Ddr1^f/f-4OHT^* and OKOΔ*Ddr1* mouse femurs (Figure 4B). Upon performing H&E staining of the femur (Figure 4C), we quantitated the cortical thickness and observed a decrease in 2- and 4-week-old OKOΔ*Ddr1* mice compared with that in age-matched *Ddr1^f/f-4OHT^* mice (Figure 4D). Moreover, quantitation of the number of mature osteoblasts in cortical bone revealed fewer cells in 2- and 4-week-old OKOΔ*Ddr1* mouse tibias than in the tibias of *Ddr1^f/f-4OHT^* mice (Figure 4C). These results indicated that osteoblasts with *Ddr1* knockout exerted decreased cortical and trabecular bone formation due to a decrease in osteoblast number, which ultimately resulted in delayed bone ossification; however, osteoblasts with *Ddr1* knockout had no influence on the growth plate length, which means that *Ddr1* knockout in osteoblasts may not affect the function of chondrocyte.

### 2.5. Knocking out Ddr1 in Osteoblasts Caused Downregulation of the Osteogenesis-Related Marker Gene Expression

To investigate whether *Ddr1* knockout in osteoblasts influences chondrocyte function at the chondro-osseous interface, we stained for chondrocyte terminal differentiation markers, such as type X collagen (Col-X) and matrix metalloproteinase-13 (MMP13). The IHC results showed that knocking out *Ddr1* in osteoblasts had no effect on either Col-X or MMP13 in the hypertrophic zone of the growth plate and the chondro-osseous interface (Figure 5B,C). We next detected that the gene expression of osteogenesis related markers, such as Runx2, and BMP2 was significantly lower in OKOΔ*Ddr1* mice than in *Ddr1^f/f-4OHT^* mice (Figure 5A). The Runx2 protein level was significantly decreased in 4-week-old OKOΔ*Ddr1* mouse trabeculae (Figure 5B). ALP is a hallmark of early development [16]. In OKOΔ*Ddr1* mice, the gene expression of ALP was decreased compared with that in *Ddr1^f/f-4OHT^* mice (Figure 5A). In osteogenesis, CoI-I and OC are markers of late osteoblast differentiation [17]. The gene expression of Col-I and OC was decreased by ~80% in OKOΔ*Ddr1* mice (Figure 5A). According to these results, we demonstrated that osteoblasts with *Ddr1* knockout delayed bone formation through downregulation of osteogenesis-related markers, indicating that *Ddr1* in osteoblasts plays a critical role in regulating osteoblast activity during osteogenesis.

### 2.6. DDR1 Promoted Mineralization in MC3T3-E1 Cells

Our results showed that knocking out *Ddr1* delayed the expression of osteogenesis-related markers in vivo, which ultimately resulted in delayed bone ossification. To better illustrate the role of *Ddr1* in mediating osteoblast functions during development, we performed gain- and loss-of-function assays in preosteoblast MC3T3-E1 cells transduced with different *Ddr1* lentiviruses. Mineralization capacity was decreased with *Ddr1* knockdown in MC3T3-E1 cells as indicated by Alizarin red staining (Figure 6A), and ALP activity was decreased in MC3T3-E1 cells with *Ddr1* knockdown (Figure 6B). On the other hand, overexpression of *Ddr1* in MC3T3-E1 cells increased both osteoblast mineralization (Figure 6C) and ALP activity (Figure 6D). These results indicated that *Ddr1* positively regulates the mineralization capacity in preosteoblast MC3T3-E1 cells. We also detected the protein levels of osteogenesis-related markers and revealed that the expression of Runx2, BMP2, ALP, Col-I, and OC was decreased in MC3T3-E1 cells with *Ddr1* knockdown (Figure 6E) and increased in MC3T3-E1 cells overexpressing *Ddr1* (Figure 6F). These results indicated that *Ddr1* regulates Runx2, BMP2, ALP, Col-I, and OC expression to mediate osteoblast differentiation and to promote mineralization in MC3T3-E1 cells.

### 2.7. DDR1 Regulated Osteoblast Differentiation through Phosphorylation of p38

Previous reports have shown that upon induction with BMP2, Runx2 transcriptional activity is regulated by ERK1/2 rather than by p38 MAP kinase in MC3T3-E1 cells [18]. We further investigated whether DDR1 regulates osteoblast differentiation via phosphorylation of p38 or ERK1/2 activity, and we detected the levels of phosphorylated p38 in MC3T3-E1 cells. The results showed that *Ddr1* knockdown decreased p38 phosphorylation but did not affect ERK1/2 phosphorylation in MC3T3-E1 cells (Figure 7A). Additionally, inhibition of p38 phosphorylation by SB203580 decreased the expression of markers related to osteoblast differentiation, such as OC, ALP, and collagen, in MC3T3-E1 cells [19]. DDR2 regulates osteoblast differentiation by regulating p38 MAPK and inducing activation of Runx2 and the extracellular matrix protein OCN [20]. To determine whether phosphorylated p38 mediated Runx2, OC, ALP, and Col-I expression, we conducted gene expression and Western blotting and showed that inhibition of p38 phosphorylation significantly decreased the gene expression of Runx2, ALP, Col-I and OC, (Figure 7B) and also the decrease in the protein levels of Runx2, ALP, Col-I and OC in MC3T3-E1 cells (Figure 7C). To confirm that *Ddr1* regulated osteogenesis of osteoblasts via phosphorylation of p38, we overexpressed *Ddr1* in MC3T3-E1 cells treated with the p38 inhibitor SB203580. The results showed that inhibiting p38 phosphorylation in MC3T3-E1 cells overexpressing *Ddr1* decreased the protein level of Runx2 (Figure 7D). These results indicated that *Ddr1* mediates Runx2 via p38 MAPK to regulate osteoblast differentiation and results in bone ossification.

## 3. Discussion

To investigate the role of *Ddr1* in osteoblasts, we created an inducible osteoblast-specific *Ddr1* knockout (OKOΔ*Ddr1*) mouse model and evaluated the roles of DDR1 during osteogenesis in vivo. Although osteoblasts with *Ddr1* knockout had no effect on mouse survival, they did show markedly reduced postnatal bone formation. Bone ossification was observed to be impaired in 2- and 4-week-old OKOΔ*Ddr1* mice. Knocking out of DDR1 in osteoblasts reduced the expression of osteogenic-related markers, including Runx2, BMP2, ALP, Col-I, and OC; these results showed reduction effects in bone mass. By using overexpression/knockdown *Ddr1* in osteoblasts in vitro, we also confirmed that *Ddr1* is required for the phosphorylation of p38 to regulate the expression of the osteogenesis-related markers Runx2, BMP2, ALP, Col-I, and OC in MC3T3-E1 cells. These results indicated that DDR1 is critical for the regulation of osteoblasts during postnatal bone formation in mice.

Osteoprogenitor cells that commit to osteoblasts eventually become mature osteocytes. Runx2 is an essential transcriptional regulator during osteoblast differentiation, and MSCs differentiate into osteoprogenitor cells with high Runx2 expression [3,21]. Cell proliferation was decreased in calvarial osteoblasts in Runx2^ΔC/ΔC^ mice [22], and in Runx2-null mice, chondrocyte and osteoblast maturation is delayed, resulting in decreased bone formation [14]. The specific deletion of exon 8 of Runx2 in osteoblasts decreased bone formation and caused osteopenia in mice [23], and DDR1 deletion reduced Runx2 activity and decreased vascular calcification [24]. p38 MAP kinase has been established as a critical contributor to early osteoblast differentiation [25]. Mice with global p38alpha knockout die in utero due to defective placental development [26]. Another study reported that induction of BMP2/7 through the ERK pathway rather than the p38 pathway increased Runx2 phosphorylation and transcriptional activity in MC3T3-E1 cells [18]. The MKK3/6-p38 pathway induces phosphorylation of Runx2 to promote osteogenesis of osteoblasts [25]. DDR1-null mice showed upregulation of p42/44 ERK and p38 to increase mesangial cell proliferation [27]. However, the role of *Ddr1* in osteoblasts remains undefined. In this study, we demonstrated that knocking out *Ddr1* in osteoblasts in vivo and knocking down *Ddr1* in MC3T3-E1 preosteoblasts in vitro decreased the expression of osteogenesis markers, such as Runx2, BMP2, ALP, Col-I, and OC. Overexpression experiments showed that DDR1 promoted osteogenesis of osteoblasts via p38 to induce Runx2 protein expression. First, we demonstrated that DDR1 regulates p38 in osteoblasts by knocking down *Ddr1* in MC3T3-E1 cells and observing no effect on ERK phosphorylation but downregulation of p38 activity. These results indicated that DDR1 regulates osteogenesis through the p38 MAPK pathway but not the ERK pathway. However, the mechanism by which *Ddr1* acts through p38 to regulate Runx2 protein levels in osteoblasts is still unknown. GATA4 (a zinc finger transcription factor) promotes Runx2 gene expression to stimulate bone mineralization and regulates ALP activity during osteoblast differentiation [28]. Knocking down GATA4 resulted in decreased levels of phosphorylated p38 and disruptions in osteoblast differentiation and bone remodeling [29]. Taken together, these data have led us to propose that *Ddr1* may act through GATA4 via p38 to regulate Runx2 activity and induce the expression of downstream markers related to osteogenesis during osteoblast differentiation and mineralization.

In the intramembranous ossification process, the bone collar from the perichondrium becomes the periosteum, which differentiates into osteoblasts. In the calcification zone in the growth plate, osteoprogenitor cells enter the area and secrete bone matrix as a scaffold to form trabecular bone. *Ddr1* is expressed at the periosteal collar of the clavicle and is especially dense in ribs in E18.5 mice [10]. In this study, the cortical and trabecular bones of OKOΔ*Ddr1* mice were significantly smaller than those of *Ddr1^f/f-4OHT^* mice at 4 weeks of age. Additionally, the cell numbers of osteoblasts were reduced in cortical bone from OKOΔ*Ddr1* mice. The difference in cortical bone thickness between *Ddr1^f/f-4OHT^* and OKOΔ*Ddr1* mice was due to a decrease in osteoblast activity, which caused a delay in ossification in the cortical and trabecular bone. Although the BMD had no significant differences between control and OKO mice, the thickness of *Ddr1*^f/f-4OHT^ was thicker than in OKOΔ*Ddr1* mice, and the BMC was significantly decreased in OKOΔ*Ddr1* mice. It showed that although BMD had not altered in OKOΔ*Ddr1* mice, the mouse size was smaller than the wild-type which caused the BMC decrease in OKOΔ*Ddr1* mice. Our results showed that the expression levels of chondrocyte terminal differentiation markers such as Col-X and MMP13 in the hypertrophic zone of the growth plate were not significantly different in the OKOΔ*Ddr1* mice compared to the *Ddr1^f/f-4OHT^* mice, which indicated that knocking out *Ddr1* in osteoblasts had no influence on the function of the chondrocytes.

It was reported that the osteoblasts had a specific enhancer of the Col-I gene in mice [30,31], and they generated a triple mutant with 2.3-kb. Col-I Cre promoter mice have time- and tissue-specific in osteoblasts and odontoblasts [32]. The Col-I Cre mRNA was firstly expressed at E13.5 in embryos which was the same with the expression of Col-I in the differentiation of osteoblasts and bone formation. The Cre recombinase specific was expressed at the cranial bones, facial bones, ribs, molars and limbs but not expressed at muscle, brain, chondrocyte in growth plate, skin, liver and kidney which also express Col-I. Thus Col-I cre mice have the most properties for our experiment.

## 4. Materials and Methods

### 4.1. Osteoblast-Specific (a1(I) Collagen-CreERT; Ddr1^f/f^) Ddr1-Deficient OKO Mice

All animal experiments were approved by the Kaohsiung Medical University Animal Care and Use Committee (IACUC105127, 1 August 2017). In our previous work, we generated conditional *Ddr1* floxed/floxed mice (*Ddr1*^f/f^) [33]. The inducible a1(I)-collagen-CreERT [B6.Cg.Tg(Col1a1-Cre/ERT2)1Crm/J] cassette contains a 2.3 kb fragment of the Col1a1 promoter, Cre recombinase, ERT2, and polyA and was purchased from Jackson Laboratories. Next, a1(I)-CreERT and *Ddr1^f/f-4OHT^* mice were crossed to generate a1(I)-collagen-CreERT, -*Ddr1*^f/f^ (OKO) mice. Genomic DNA from tail tips was extracted, and gene expression was confirmed by polymerase chain reaction (PCR). The presence of the 3′ loxP and Cre sites was verified by PCR using the following primers: loxP, 5′-ATAGCGGCCGCTGCTGGTCTTAGCTCTGT-3′ and 5′-ATAGTCGACACAGAGAGTTAAGCCAGA-3′; Cre, 5′-TTCAATTTACTGACCGTACACCAA-3′ and 5′-CCTGATCCTGGCAATTTCGGCTA-3′.

### 4.2. Induction of 4-Hydroxytamoxifen (4-OHT)

In the Cre-LoxP system, activation of Cre recombinase requires 4-hydroxytamoxifen (4-OHT). The 4-OHT was dissolved in DMSO to generate a stock solution at a concentration of 25 mg/mL, and the working concentration was 4 mg/kg diluted in corn oil (C8267, Sigma-Aldrich, St. Louis, MO, USA) at an oil:4-OHT ratio of 9:1. This 4-OHT solution was intraperitoneally injected into E14.5 mice (4 mg/kg/day) along with progesterone (2 mg/day/kg; P0130, Sigma-Aldrich). After birth, 1 mg/kg 4-OHT was injected per day for 5 consecutive days. Then, 2- and 4-week-old mice were collected. Both *Ddr1*
^f/f-4OHT^ (control group) and OKOΔ*Ddr1* (experimental group) mice were injected with 4-OHT on the abovementioned schedule.

### 4.3. Double Staining Analysis for Skeleton

Two- and 4-week-old *Ddr1 ^f/f-4OHT^* and OKOΔ*Ddr1* mice were harvested, and their skeletons were fixed in 4% paraformaldehyde for 24 h followed by soaking in freshly prepared 2% Alcian blue 8GX (A5268, Sigma-Aldrich) in 1% acetic acid for 2 days. Then, the skeletons were washed in 0.5% KOH (60377, Sigma-Aldrich) in 1X PBS until the muscles became transparent. Finally, the skeletons were stained with freshly prepared 1% Alizarin red S (A5533, Sigma-Aldrich) and for 10 min and washed with 0.5% KOH to removed excess dye. The skeletons were observed under a Leica-DSM1000 microscope (Leica Microsystems, Wetzlar, Germany).

### 4.4. Microcomputed Tomography (Micro-CT)

High-resolution microcomputed tomography (micro-CT) analysis (SkyScan 1076; SkyScan NV, Kontich, Belgium) was used to render 3-D reconstructions of 4-week-old mouse tibias with the following conditions: isotropic voxel resolution, 9 μm; 0.5 mm aluminum filter; tube voltage, 50 kV; tube current, 200 μA; and exposure time, 1100 ms. A scale of 0–0.09 (NRecon version 1.6.1.7; SkyScan NV) was used to reconstruct the 3-D morphometric parameters of 5 mm of the tibia. For the trabecular analysis, the inner circle of the ROI (2 mm circle; 120 cuts) beneath the growth plate from 0.5 mm to 2.5 mm (total length of 2 mm) was selected for analysis of the trabecular bone mineral density (BMD, g/cm^3^), percent bone volume (BV/TV, %), trabecular thickness (Tb.Th, µm), trabecular number (Tb.N, mm) and trabecular separation (Tb.Sp, µm). A total length 2 mm outer circle of the ROI at the middle of the tibia was selected for cortical analysis of the cortical bone volume (BV, mm^3^), bone mineral content (BMC, mg/mm), and bone mineral density (BMD g/cm^3^).

### 4.5. Three-Point Bending Test

The 4-week-old mouse femurs were subjected to 3-point bending and tested to failure by using a materials testing system (INSTRON 5943, Norwood, MA, USA). The femur was placed on 2 lower supports 2 mm apart. The load was placed at the middle of the femur at a displacement rate of 0.05 mm/s until failure. Quantitative analysis was performed with Bluehill software (Illinois Tool Works Inc., Glenview, IL, USA).

### 4.6. Histological and Immunohistochemistry (IHC) Staining

The femur was harvested, fixed in 4% PFA for 3 days, demineralized with 0.5 M EDTA, and embedded in paraffin. Sections 5 µm thick were sliced and then rehydrated for histological staining as follows: hematoxylin and eosin staining, hematoxylin (MHS1, Sigma-Aldrich, St. Louis, MO, USA) for 120 sec, a ddH2O wash, and eosin (230251, Sigma-Aldrich) for 3 sec; and Safranin O/fast green staining, hematoxylin (MHS1, Sigma-Aldrich) for 180 s, a ddH2O wash, 0.05% fast green (2353-45-9, Sigma-Aldrich) for 90 s, and 0.1% Safranin O red (HT90432, Sigma-Aldrich). For IHC staining, antigen retrieval was conducted by boiling the sections in 0.5% tris-EDTA in 1X PBS for 30 min. Next, the sections were incubated with 3% H_2_O_2_ peroxidase for 10 min at room temperature, blocked with blocking buffer (ab126587, Abcam, Cambridge, MA, USA) and incubated overnight at 4 °C with primary antibodies targeting DDR1 (PA5-29316, Thermo Fisher Scientific Inc., Waltham, MA, USA), DDR2 (GTX102526, Gene Tex Inc., Irvine, CA, USA), BMP2 (ab14933, Abcam), ALP (ab83259, Abcam), Col-I (ab34710, Abcam), Runx2 (ORB10256, Biorbyt LLC Ltd., San Francisco, CA, USA), and osteocalcin (GTX13418, Gene Tex Inc., Irvine, CA, USA). The sections were then treated with mouse- and rabbit-specific HRP/DAB detection kits (ab64264, Abcam), counterstained with hematoxylin and observed under a Leica-DM1750 microscope (Leica Microsystems, Wetzlar, Germany).

### 4.7. Cell Culture and Drug Treatment

The mouse preosteoblast cell line MC3T3-E1 (CRL-2593) (ATCC, Manassas, VA, USA) was grown in α-minimal essential (α-MEM) medium supplemented with 10% fetal bovine serum (FBS) and 100 units/mL penicillin and was maintained at 37 °C with 5% CO_2_. SB203580 (Ciliobrevin A, Selleckchem, Houston, TX, USA), an inhibitor of phosphorylated p38, was dissolved in DMSO as a stock solution, and the cells were treated with the inhibitor at a final concentration of 20 μM for 16 h, after which the cells were harvested for Western blot analysis.

### 4.8. Lentivirus Constructs and Transfection

Lentivirus was used to transiently knock down or overexpress *Ddr1* in MC3T3-E1 cells. Purchased lentivirus particles with *shLacZ* (as control), *shDdr1* (as knockdown *Ddr1*, TRCN0000010084), vehicle (as a control for *Ddr1* overexpression, PLAS2W.Pp), and ov*Ddr1* (as overexpress *Ddr1*, PLAS2W.DDR1.Pp) from the RNAi Core Facility of Taiwan. MC3T3-E1 cells were cultured in 10 cm dishes at a density of 1 × 10^6^, infected lentivirus particles for 16 h (MOI of 1) and used puromycin (2 μg/mL) to select stable cell lines.

### 4.9. Real-Time PCR (qPCR) Analysis

MC3T3-E1 cells were treated with SB203580 for 24 hr and the in vivo calvarial bone was extracted from postnatal 4- to 5-day-old *Ddr1^f/f-4OHT^* and OKOΔ*Ddr1* mice and placed on ice. The PureLink RNA mini kit (Thermo Fisher Scientific Inc., Waltham, MA, USA.) was used to obtain total RNA, and then cDNA was transcribed from 2 µg of total RNA with the Verso 1-Step RT-PCR Kit ReadyMix with ThermoPrime Taq. For quantitative real-time PCR, 6.25 uL of SYBR (BIO-RAD Laboratories Inc., Hercules, CA, USA), 1 uL of cDNA, 0.5 uL of specific primers (sequences available in Appendix A), and ddH_2_O to a total volume of 13 uL were mixed and run on a CFX connect real-time PCR detection system (BIO-RAD Laboratories Inc., Hercules, CA, USA). GAPDH was used as a housekeeping gene, and expression was normalized to that in *Ddr1^f/f-4OHT^* mice.

### 4.10. Western Blot Analysis

Primary cells were extracted from the flat bone of P3 *Ddr1^f/f-4OHT^* and OKOΔ*Ddr1* mice and MC3T3-E1 cells were extracted with RIPA extraction buffer (89900, Thermo Fisher Scientific Inc., Waltham, MA, USA.) containing 1% proteinase inhibitor (78429, Thermo Fisher Scientific Inc., Waltham, MA, USA) and 1% phosphatase inhibitor (78426, Thermo Fisher Scientific Inc., Waltham, MA, USA). Western blot was performed with primary antibodies against DDR1 (ab22719, Abcam, Cambridge, MA, USA), collagen type I (ab34710, Abcam), collagen type X (ab58632, Abcam), GAPDH (MA5-15738, Thermo Fisher Scientific Inc.), ALP (ab95462, Abcam, Cambridge, MA, USA), Runx2 (ab23981, Abcam), BMP2 (ab14933, Abcam), Col-I (ab34710, Abcam), Osteocalcin (GTX13418, Gene Tex Inc., Irvine, CA, USA), phosphorylated 44/42 ERK1/2 (D11A8, Cell Signaling Inc., Danvers, MA, USA), ERK1/2 (137F5, Cell Signaling Inc.), phosphorylated p38 (Thr180/Tyr182) (ARG51580; Arigo biolaboratories, Hsinchu, TW), p38 (ARG55258; Arigo biolaboratories) and mouse and rabbit secondary antibodies (615005214 and 611005215; Jackson immune research; West Grove, PA, USA). Protein bands were detected by enhanced chemiluminescence analysis (ECL system; GE Healthcare, Piscataway, NJ, USA).

### 4.11. Mineralization Assay

To generate stable cell lines, cells were transduced with *Ddr1* lentivirus particles, subcultured and seeded at a density of 1 × 10^4^ cell/cm^2^ in 48-well plates. Once cells reached 80% confluence, the medium was changed to induction medium (α-MEM with 10% FBS, 100 unit/mL penicillin/streptomycin, 50 μM ascorbic acid, and 5 mM β-glycerophosphate) containing 1 mg/mL puromycin. The selective induction medium was replaced every 2 days for 21 days. Next, the cells were washed in PBS, fixed with 10% formaldehyde for 10 min, washed with ddH2O at least 5 times and stained with 2% Alizarin red (A5533, Sigma-Aldrich) solution (pH 4.1) for 5 min. After staining, the cells were washed in ddH2O and imaged with a camera. Quantitation of the staining intensity was performed by adding 200 µL of 10% acetic acid per well and shaking the plates for 10 min, after which 100 uL of the solution was added to 96-well plates per well and the OD415 was measured with a TECAN Sunrise ELISA reader (Switzerland) (*n* = 6).

### 4.12. Alkaline Phosphatase Assay

MC3T3-E1 cells were transduced with *Ddr1* lentivirus and seeded at a concentration of 1*10^4^ in 48-well plates. Next, induction medium with 1 mg/mL puromycin was replaced every 2 days for 21 days. An ALP assay kit (86R, Sigma-Aldrich) was used according to the manufacturer’s instructions. The culture medium was removed, and the cells were washed once in PBS. Lysate buffer was added to the suspension, which was then centrifuged at 12,000 rpm. The pellet was resuspended and transferred to 96-well plates, to which ALP substrate solution (50 uL/per well) was added. The plates were incubated at 37 °C with a 5% CO_2_ humidified atmosphere after which the absorbance at 405 nm was detected by a TECAN Sunrise ELISA reader (Switzerland) (*n* = 6).

### 4.13. Statistical Analysis

All animal experiments are expressed as the mean (SEM) of 6 independent groups. Each condition was repeated at least three times, and the total number of mice per group was more than 6. Statistical significance was determined using Student’s *t*-test, and multiple comparisons were performed using Scheffé’s method. In this study, the groups were compared with the *Ddr1^f/f-4OHT^* group. For the *p* values, (*) indicated *p* < 0.05, and (**) indicated *p* < 0.01, and (***) indicated *p* < 0.001.

## 5. Conclusions

In conclusion, by using overexpression and knockdown of Ddr1 in osteoblasts, we demonstrated that DDR1 in osteoblasts regulates osteoblast differentiation via phosphorylation of p38 to regulate the expression of Runx2 and downstream markers related to osteogenesis, such as Runx2, BMP2, ALP, Col-I, and OC. This mechanism results in the reduction in long bone ossification and weakened mechanical properties in OKOΔ*Ddr1* mice in vivo, which indicate that *Ddr1* plays a crucial role in osteoblast function to regulate the differentiation of osteoblasts. Our findings may be applied to patients with osteoporosis or fracture as a basis for developing *Ddr1*-targeted therapy.

## Figures and Tables

**Figure 1 ijms-21-07210-f001:**
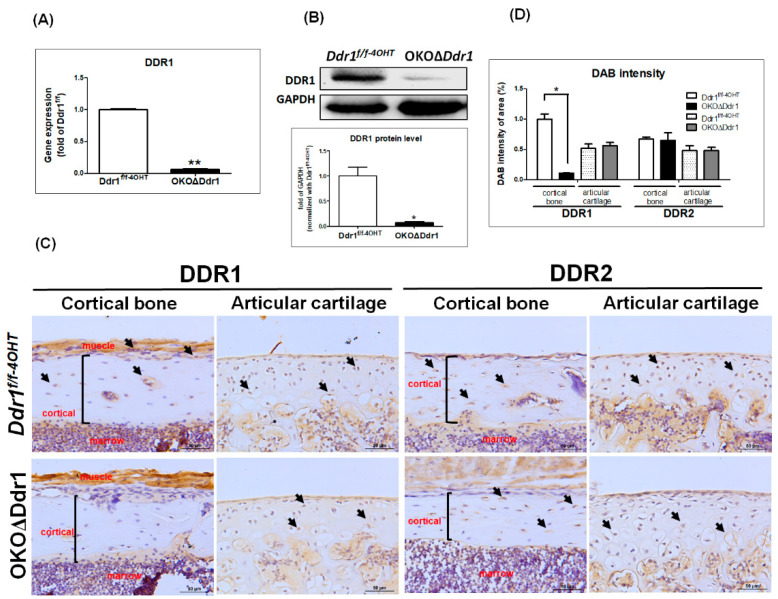
Generation of osteoblast-specific (a1(I) collagen-CreERT; *Ddr1*^f/f^) OKOΔ*Ddr1* mice. The calvarial bone was extracted from *Ddr1^f/f-4OHT^* and OKOΔ*Ddr1* mice on postnatal days 4~5, and (**A**) gene expression of *Ddr1* and (**B**) protein level of DDR1 were detected. The ratio of DDR1/GPADH is expressed relative to the *Ddr1^f/f-4OHT^* mice was quantified. Each group *n* ≥ 6. (**C**) IHC staining of DDR1 and DDR2 in cortical bone and articular cartilage of femurs from 4-week-old mice; black arrow indicates DDR1-positive cells; black frame indicates the cortical region. (**D**) Quantitative results by tissue faxes normalized with *Ddr1^f/f-4OHT^*. Magnifications of 400X are shown, with scale bars of 50 μm. Each group, *n* ≥ 6; * *p* ≤ 0.05, ** *p* ≤ 0.01.

**Figure 2 ijms-21-07210-f002:**
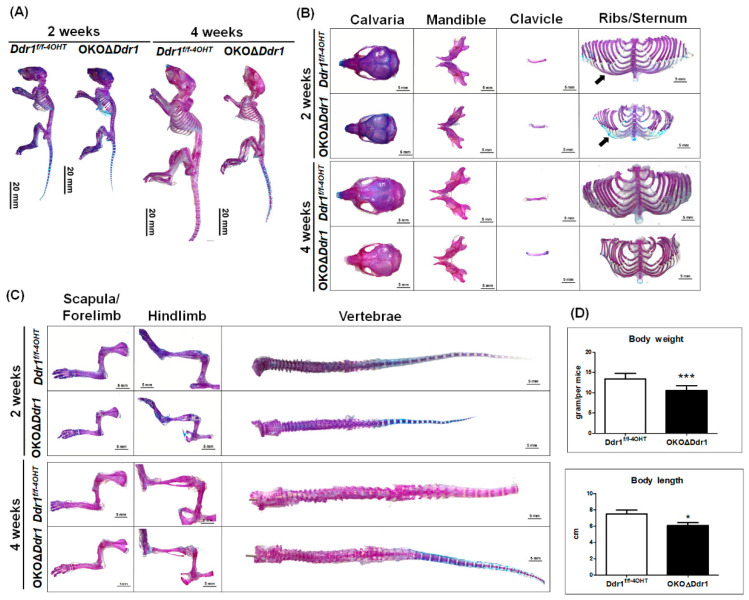
Skeletal dysplasia in OKOΔDdr1 mice. (**A**) Double staining of skeletons from 2- and 4-week-old *Ddr1^f/f-4OHT^* and OKOΔ*Ddr1* mice. Anatomy microscopy image (0.75×) with scale bar 20 mm. (**B**) Intramembranous ossification bone, including calvaria, mandible, clavicle, and ribs/sternum. The image was obtained at 1.25× magnification with a scale bar of 5 mm. The black arrow indicates rib cartilage of ribs. (**C**) Endochondral ossification bone including scapula/forelimb, hindlimb, and vertebrae. The anatomy microscopy image was 1.25× with a scale bar of 5 mm. (**D**) Body weight and body length were assessed in 4-week-old mice. Each group, *n* ≥ 6; * *p* ≤ 0.05, *** *p* ≤ 0.001.

**Figure 3 ijms-21-07210-f003:**
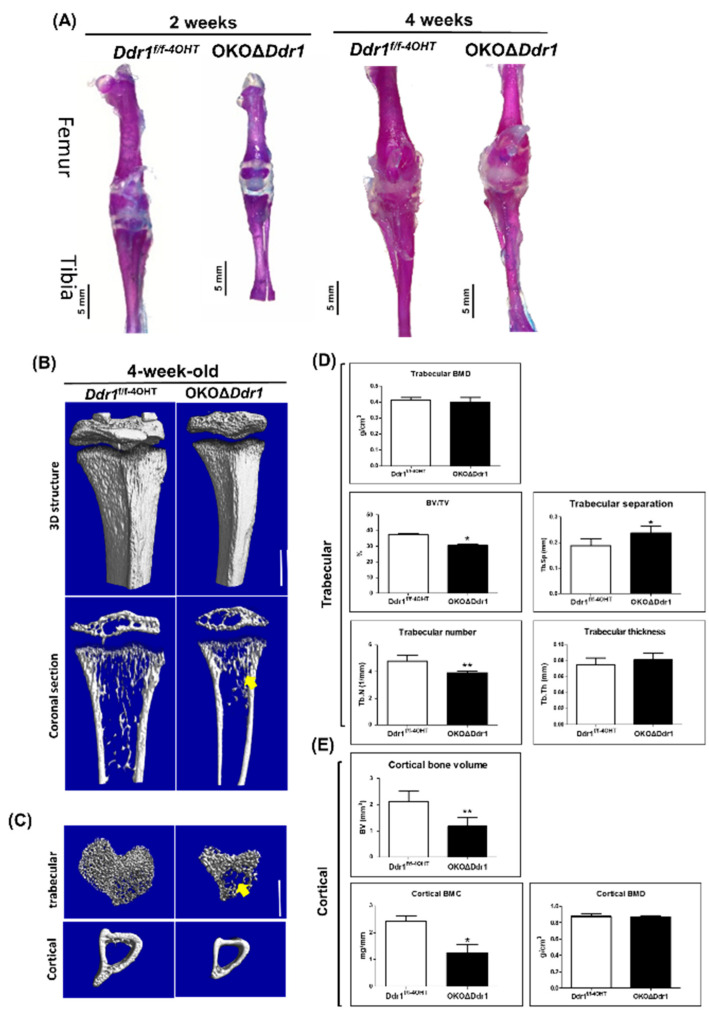
Decreased bone formation in the tibias of OKOΔDdr1 mice. (**A**) Double staining of hindlimb bones (femur and tibia) from 2- and 4-week-old mice. Microscopy image (1.25× magnification) of the hindlimbs; scale bar 5 mm. Top is the femur, and bottom is the tibia. The micro-CT reconstructed 3D structure of (**B**) the total length with 5 mm of tibia, the coronal section view of the tibia, (**C**) and reconstructed 2 mm 3D structure of trabecular bone and cortical bone in *Ddr1^f/f-4OHT^* and OKOΔ*Ddr1* mice with a scale bar of 1 mm. The yellow arrow indicates trabecular bone. (**D**) Quantitation of trabecular bone mineral density (BMD), trabecular bone volume/total volume (BV/TV), trabecular number (Tb.N), trabecular separation (Tb.Sp), and trabecular thickness (Tb.Th) (**E**) Quantitation of cortical bone volume, cortical bone mineral density (BMD), and cortical bone mineral content (BMC) with total 2 mm at middle of tibia. Each group *n* ≥ 6; * *p* ≤ 0.05, ** *p* ≤ 0.01.

**Figure 4 ijms-21-07210-f004:**
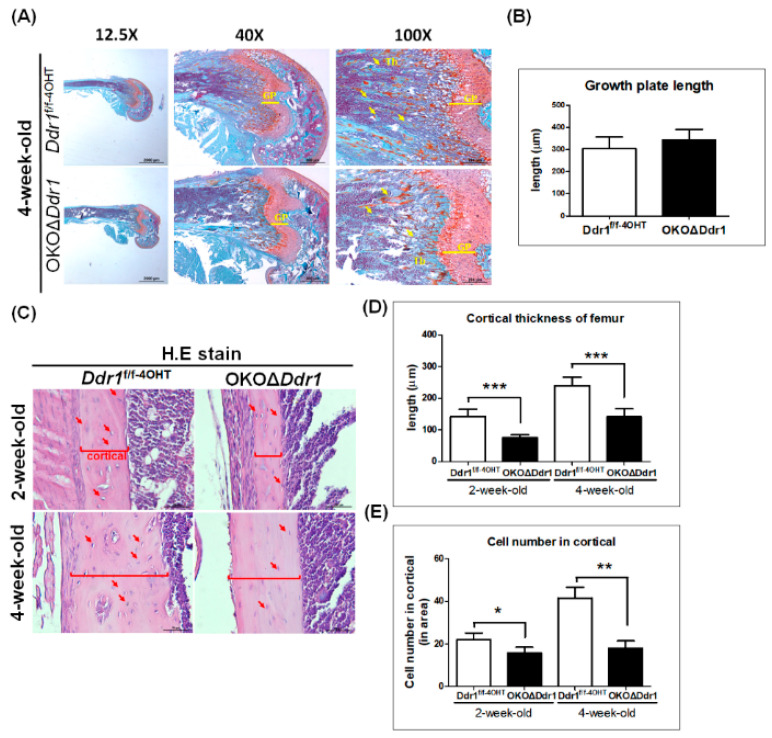
Knocking out Ddr1 in osteoblasts decreases cortical bone thickness and cell number. (**A**) Safranin O/fast green staining of femurs from 4-week-old *Ddr1^f/f-4OHT^* and OKOΔ*Ddr1* mice. Magnifications of 12.5×, 40×, and 100× are shown, with scale bars of 2000 μm, 500 μm, and 200 μm, respectively. GP indicates growth plate; yellow arrow indicates trabecular bone. (**B**) Quantitation of the growth plate length. (**C**) H&E staining of femur with cortical bone. The red arrow indicates osteoblasts; the red frame indicates cortical bone thickness. Quantitation of the (**D**) cortical thickness and (**E**) cell number in cortical bone per 0.12 mm^2^ in 2- and 4-week-old mouse femurs. Each group *n* ≥ 6; * *p* ≤ 0.05, ** *p* ≤ 0.01, *** *p* ≤ 0.001.

**Figure 5 ijms-21-07210-f005:**
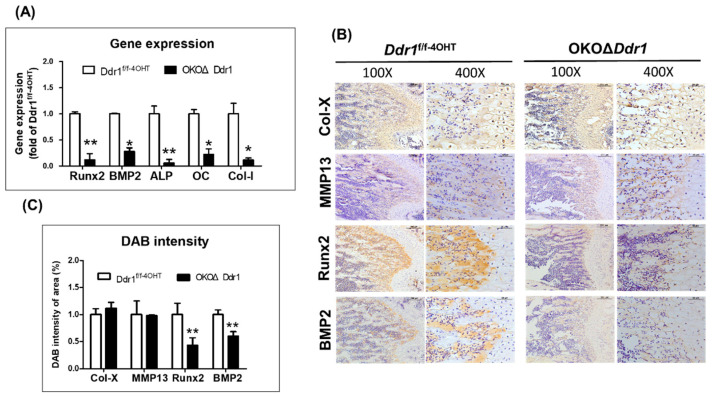
Knocking out *Ddr1* in osteoblasts caused downregulation of the gene expression of markers related to osteogenesis. (**A**) Gene expression of Runx2, BMP2, ALP, OC, and Col-I was assessed in calvarial bone of *Ddr1^f/f-4OHT^* and OKOΔ*Ddr1* mice. (**B**) IHC staining of Col-X, MMP13, Runx2, and BMP2 in femurs from 4-week-old *Ddr1^f/f-4OHT^* and OKOΔ*Ddr1* mice. Magnifications were 100× and 400× with scale bars of 200 μm and 50 μm, respectively. (**C**) Quantitation of Col-X, MMP13, Runx2, and BMP2 expression in the trabecular area. Each group *n* ≥ 6; * *p* ≤ 0.05, ** *p* ≤ 0.01.

**Figure 6 ijms-21-07210-f006:**
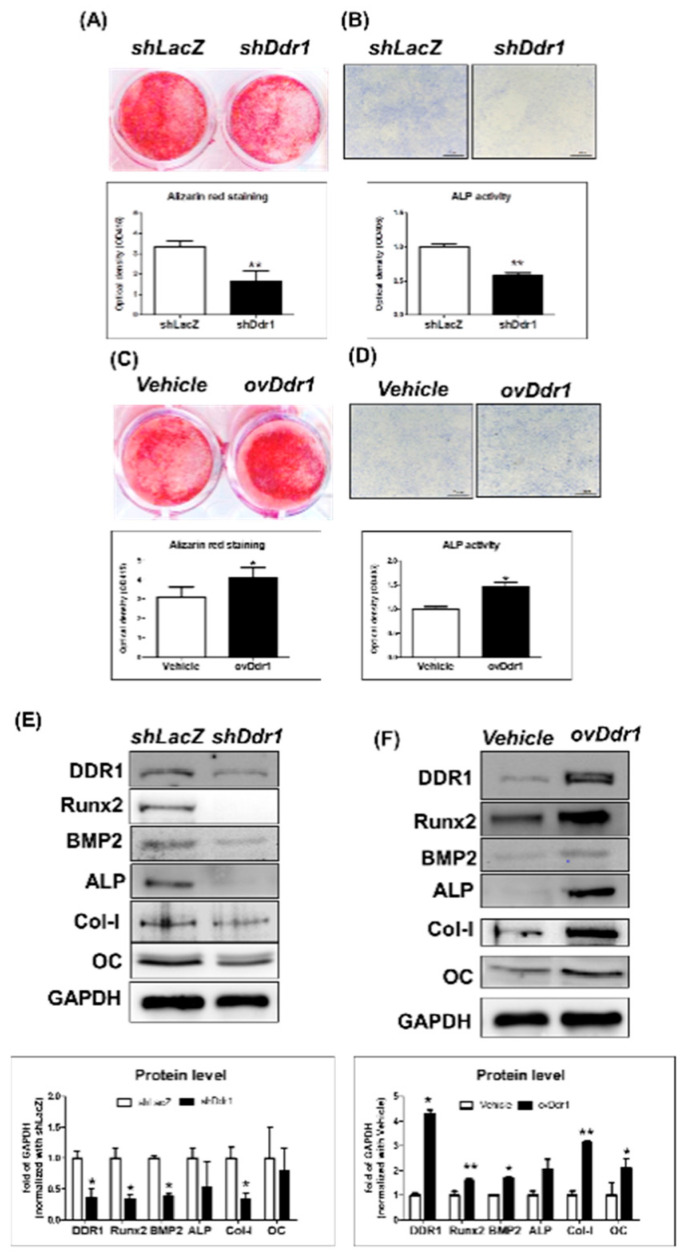
DDR1 promoted mineralization in MC3T3-E1 cells. Knocking down *Ddr1* (*shDdr1*) in ME3T3-E1 cells was performed with (**A**) mineralization assay staining by Alizarin red and (**B**) ALP activity. The quantitative results are shown below. (**C**) The protein levels of DDR1, Runx2, BMP2, ALP, Col-I, OC, and GAPDH in ME3T3-E1 cells were assessed. Quantitation of the results for *shDdr1* compared with *shLacZ* and all other results is shown below. ME3T3-E1 cells overexpressing *Ddr1* (*ovDdr1*) were subjected to (**D**) the mineralization assay staining by Alizarin red and (**E**) ALP activity. The quantitative results are shown below. (**F**) The protein levels of DDR1, Runx2, BMP2, ALP, Col-I, OC, and GAPDH were assessed in ME3T3-E1 cells. Quantitation of *ovDdr1* expression compared with *shLacZ* overexpression is shown below. Each group included *n* ≥ 3 independent groups; * *p* ≤ 0.05, ** *p* ≤ 0.01.

**Figure 7 ijms-21-07210-f007:**
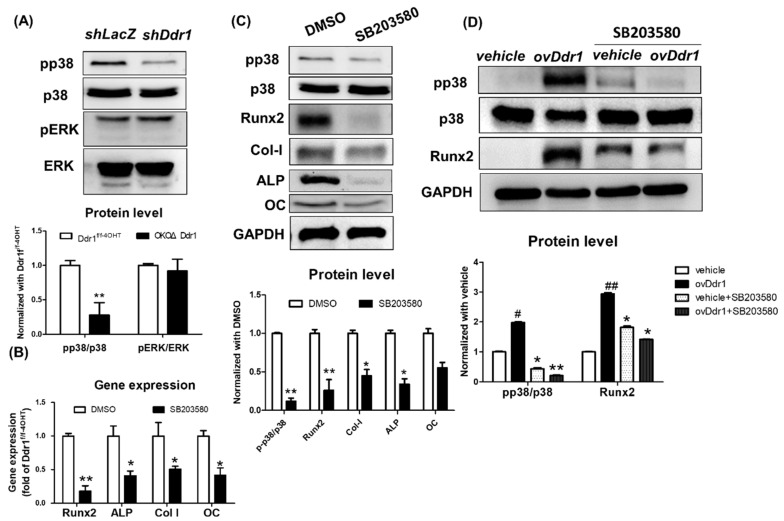
DDR1 regulated osteoblast differentiation through p38 phosphorylation. (**A**) The protein levels of phosphorylated p38 and ERK in ME3T3-E1 cells with Ddr1 knockdown (*shDdr1*). The quantitative result was presented as the pp38/p38 and pERK/ERK ratios and normalized with *shLacZ*. (**B**) Gene expression of Runx2, ALP, Col-I, OC in ME3T3-E1 cells were treated with the p38 inhibitor. (**C**) Western blot detected the protein level of phosphorylated p38, Runx2, Col-I, ALP, and OC Runx2 in MC3T3-E1 treated with DMSO (control) and SB203580 (p38 inhibitor). The quantitative results show the ratios of pp38/p38 and Runx2/GAPDH, Col-I/GAPDH, ALP/GAPDH, and OC/GAPDH and were normalized with *DMSO.* (**D**) The protein levels of Runx2, pp38, p38 and GAPDH in ME3T3-E1 cells overexpressing *Ddr1* (*ovDdr1*) and treated with SB203580. Quantitation of the results of *ovDdr1+DMSO* compared with *vehicle+DMSO and of ovDdr1+SB203580* compared with *vehicle+SB203580* and normalized with those of vehicle+DMSO is shown below. Each group included *n* ≥ 3; * *p* ≤ 0.05, ^#^
*p* ≤ 0.05, ** *p* ≤ 0.01, ^##^
*p* ≤ 0.01 and * means compare with *shLacZ* or DMSO or *vehicle,* and # mean compare with *ovDdr1*.

**Table 1 ijms-21-07210-t001:** Decreased mechanical properties in the femur of OKOΔDdr1 mice.

Group		Ddr1^f/f-4OHT^	(Mean ± SE)	OKO△Ddr1	(Mean ± SE)	*p* Value	
MaxLoad	N	10.68	±3.55	4.26	±0	0.001	**
Break point	N	8.31	±3.42	4.04	±0.01	0.01	*
stiffness(S)	N/mm^2	70.69	±31.13	33.51	±0.01	0.006	**
area under the Max curve(AUC1)	J	0.003	±0.0017	0.0012	±0.0099	0.01	*
area under the Break curve(AUC2)	J	0.0059	±0.0031	0.0028	±0.0134	0.013	*
UStress	MPa	0.53	±0.23	0.37	±0.19	0.188	
Modulus	GPa	3.23	±0.91	1.83	±0.14	0.048	*
Toughness	MPa	1.77	±0.73	0.94	±0.059	0.04	*

Used the three-point bending analysis the 4-week-old *Ddr1^f/f-4OHT^* mice and OKOΔ*Ddr1* mice with max-load, break point, stiffness, area under the max curve, area under the break curve, ustress, modulus, and toughness in femur. Each group *n* ≥ 6; * *p* ≤ 0.05, ** *p* ≤ 0.01. *p* value: compare with Ddr1f/f-4OHT.

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
