# Peer review of "Discoidin Domain Receptor 1 Regulates Runx2 during Osteogenesis of Osteoblasts and Promotes Bone Ossification via Phosphorylation of p38"

_ijms, 2020, doi:10.3390/ijms21197210_

Round 1

Reviewer 1 Report

The authors answered my main concerns.

Author Response

Thank you for your comment.

Reviewer 2 Report

Although the paper has been partially improved, there remained still several serious concerns in this paper.

  1. Although they explained the reason of the use of type 1 collage-Cre mice, the phrase, #“Osterix were expressed at perichondrium, pre-hypertrophic, hypertrophic chondrocytes, which promoted chondrocyte’s maturation; and also expressed at pre-osteoblasts, which promote osteoblast’s maturation, differentiation, lead to secrete osteocalcin. The osteocalcin was late osteoblast marker and which was described as an endocrine hormone. Therefore, even osterix-cre and costecalcin-cre have been used for osteoblast specific gene knockout mice, it wasn’t suitable in our study.” is not appropriate for the explanation. At least, these sentences should be deleted.
  2. The merit of this paper is the analysis of osteoblast-specific DDR2 knockout mice, and the significance of osteoblast phenotype cannot be shown in the absence of the presentation of some osteoclast phenotype. At least, TRAP stain can be performed easily and quickly. The data should be shown in this paper. The description of the future study is not acceptable.
  3. Re: Figure 5C, the data of ALP, Col-1 and osteocalcin were not reliable and not compatible with gene expression data. The specificity of those antibodies are usually bad. Therefore, Figure 5 B and C should be deleted.”

The experimental quality of Figures 5 is not scientifically acceptable. This will reduce the reliability of whole paper, if the authors wish the presentation of the data..

  1. English of the corrected sentences in the revised version is not good. English edition by a native speaker is necessary.

Author Response

Response to Reviewer 2 Comments

Point 1: Although they explained the reason of the use of type 1 collage-Cre mice, the phrase, #“Osterix were expressed at perichondrium, pre-hypertrophic, hypertrophic chondrocytes, which promoted chondrocyte’s maturation; and also expressed at pre-osteoblasts, which promote osteoblast’s maturation, differentiation, lead to secrete osteocalcin. The osteocalcin was late osteoblast marker and which was described as an endocrine hormone. Therefore, even osterix-cre and costecalcin-cre have been used for osteoblast specific gene knockout mice, it wasn’t suitable in our study.” is not appropriate for the explanation. At least, these sentences should be deleted.

Response 1:  Thank you for your comment. We have deleted the sentences in Discussion section.

Point 2: The merit of this paper is the analysis of osteoblast-specific DDR1 knockout mice, and the significance of osteoblast phenotype cannot be shown in the absence of the presentation of some osteoclast phenotype. At least, TRAP stain can be performed easily and quickly. The data should be shown in this paper. The description of the future study is not acceptable.

Response 2: Thank you for your comment. We have stained the osteoclasts by TRAP staining, and also immunochemistry stain of TRAcP as an osteoclast’ marker. The results showed that knockout Ddr1 in osteoblasts has no effect on the osteoclasts activity and the express of osteoblasts (S1A, 1B). These results indicated that knockout Ddr1 in osteoblasts wound not influence osteoclast phenotype, demonstrate the osteoblasts specific Ddr1 knockout in mice.

 in the page 6-7 at line 24-2

Point 3: Re: Figure 5C, the data of ALP, Col-1 and osteocalcin were not reliable and not compatible with gene expression data. The specificity of those antibodies are usually bad. Therefore, Figure 5 B and C should be deleted.”

The experimental quality of Figures 5 is not scientifically acceptable. This will reduce the reliability of whole paper, if the authors wish the presentation of the data.

Response 3: Thank you for your comment. We have deleted the Figure 5 B and C in results and Figure section.

Point 4: English of the corrected sentences in the revised version is not good. English edition by a native speaker is necessary.

Response 4: Thank you for your comment. We have send the English edition to MDPI English edition ().

Round 2

Reviewer 2 Report

The authors responded to the Reviewer's comments adequately. However, some responses are not observed in the revised manuscript and figures. Please reflect all your responses and English edition to the manuscript and figures. 

This manuscript is a resubmission of an earlier submission. The following is a list of the peer review reports and author responses from that submission.

Round 1

Reviewer 1 Report

Using Ddr1 knockout mice and MC3T3-E1 cells, Chou et al. showed that DDR1 can modulate Runx2 expression through phosphorylation of p38.

The knockout mice displayed many osteogenic malformations that denote of the role of DDR1 protein in the endochondral and intramembranous ossification.

The experimental procedures were appropriate to demonstrate the role of DDR1 during osteoblastogenesis.

Overall, the data support very well the involvement of DDR1 in osteoblast function because the authors have investigated many osteogenic gene and protein markers. They demonstrate DDR1 activation leads to p38 phosphorylation that somehow activates the expression of the master gene of osteoblastogenesis Runx2. They also show that this activation is mediated through the p38 MAP kinase pathway but not through the ERK pathway. It is tempting to ask if the TGF-b pathway can contribute partially or any to this process. It is widely accepted that Runx2 can be activated through this pathway through SMAD phosphorylation. The latter was not addressed at all in the authors investigations, neither in mice nor in the MC3T3-E1 cells.

The authors have used many opposing statements about the role of DDR1 in the endochondral ossification:

p11-L50-51:

“These results suggest that Ddr1 knockout in osteoblasts disturbs the process of endochondral and intramembrane ossification. »

p6-L15:

Ddr1 knockout in osteoblasts did not affect chondrocyte function. »

p11-L46 :

“knocking out Ddr1 in osteoblasts had no effect on chondrocyte function. »

Despite the contradictory conclusions, the master gene in chondrocytes is believed to be SOX9. The expression level of this gene was not addressed in any of the biological materials used in this study and it is worth looking after.

Minors:

Some paragraphs are confusing because of the use of abbreviations that have not been previously defined.

p11-L7 : Runx2, BMP2, ALP are not bone matrix proteins. The authors should rephrase this sentence.

Reviewer 2 Report

General comments

In this study, they investigated the effects of discoidin domain receptor 1 (Ddr1) on osteoblast differentiation in vivo and in vitro using osteoblast-specific Ddr1-deleted mice.

They showed that trabecular and cortical bone volume as well as osteoblast differentiation parameters were decreased in Ddr1-deleted mice. In vitro results were compatible with in vivo data.

Although the data may be potentially useful, some data are not concrete. Moreover, there are several points, which should be addressed.

Specific comments

  1. In Introduction, the bone-related phenotypes in global Ddr1-null mice should be described. Are there any reports available about osteoblast-specific Ddr1-deleted mice? These points should be described in Introduction.
  2. Osterix- and osteocalcin-Cre mice have been usually used for osteoblast specific gene knockout mice, since type 1 collagen is not specific for osteoblasts. Why they used type 1 collage-Cre mice should be explained. Moreover, this point should be discussed in Discussion as the major limitation of the study.
  3. In figure 3, Ddr1 deletion did not affect bone mineral density, which are not compatible with the other data. This discrepancy should be described in detail in Discussion. Moreover, the clear analyses separating trabecular and cortical bone mineral density should be performed. The manuscript seemed to be difficult to understand. The data presentation should be improved.
  4. Re: Page 5, lines 9-11, the meaning of the sentence is not clear, which should be rewrite.
  5. Re: Figure 4, Osteoclasts were not evaluated. Evaluation of osteoclasts by TRAP stain should be performed. Moreover, the evaluation of osteoblast apoptosis could be useful.
  6. Re: Figure 5C, the data of ALP, Col-1 and osteocalcin were not reliable and not compatible with gene expression data. The specificity of those antibodies are usually bad. Therefore, Figure 5 B and C should be deleted. ALP statin or Osterix immune stain will be more useful.
  7. Re: Figure 7B, 7C, Western blots for Col-1, ALP and osteocalcin are not reliable. Real time PCR data of those experiments should be shown.
  8. There are several minor errors in the manuscript and figure.

Reviewer 3 Report

The article by Chou et al., discussed the DDR1 mediates the expression and activity of Runx2, and promotes bone ossification through regulation of p-38 phosphorylation. The authors evaluate the expression of osteoblast differentiation in the vivo model of osteoblast-specific Ddr1 knockout mice (OKOΔDdr1) and also used an in vitro model of Ddr1 knockdown in osteoblasts, which make the work complete and rigorous. Here please consider some comments from my side:

1, From fig 6, Author suggested that Ddr1 regulates Runx2, BMP2, ALP, Col-I, and OC expression to mediate osteoblast differentiation and to promote mineralization in MC3T3-E1 cells. The Fig 6 E, F, we can see that the expression of Runx2 and BMP2 appeared similar results. Please explain the effects of these two on mediation of osteoblasts and why Ddr1 mediates Runx2, but not BMP2, for the regulation of osteoblast maturation and bone mineralization.

2, Please check and modify the description below figures. There are incorrect descriptions below some figures in results, some descriptions do not match the labeling order of the figures.